# Serotonin signaling modulates aging-associated metabolic network integrity in response to nutrient choice in *Drosophila melanogaster*

Yang Lyu [1 ✉], Daniel E. L. Promislow[2,3] & Scott D. Pletcher [1 ✉]

Aging arises from complex interactions among multiple biochemical products. Systems-level analyses of biological networks may provide insights into the causes and consequences of aging that evade single-gene studies. We have previously found that dietary choice is sufficient to modulate aging in the vinegar fly, *Drosophila melanogaster*. Here we show that nutrient choice influenced several measures of metabolic network integrity, including connectivity, community structure, and robustness. Importantly, these effects are mediated by serotonin signaling, as a mutation in serotonin receptor 2A (*5-HT2A*) eliminated the effects of nutrient choice. Changes in network structure were associated with organism resilience and increased susceptibility to genetic perturbation. Our data suggest that the behavioral or perceptual consequences of exposure to individual macronutrients, involving serotonin signaling through 5-HT2A, qualitatively change the state of metabolic networks throughout the organism from one that is highly connected and robust to one that is fragmented, fragile, and vulnerable to perturbations.

[1] Department of Molecular and Integrative Physiology and Geriatrics Center, Biomedical Sciences and Research Building, University of Michigan, Ann Arbor, MI, USA. [2] Department of Lab Medicine and Pathology, University of Washington School of Medicine, Seattle, WA, USA. [3] Department of Biology, University of Washington, Seattle, WA, USA. ✉email: yanglyu@umich.edu; spletch@med.umich.edu

Emergent, system-level properties of biological networks have provided insights into many complex behaviors of organisms that single-gene or pathway analyses often struggle to explain[1–3]. For example, robust gene networks allow animals to maintain segment patterning against perturbations during embryonic development, through interactions and feedback controls among multiple transcriptional modules within and between cells[4]. Regulatory loops in signal transduction networks render bacterial chemotaxis immune to intrinsic noises[5]. During eye development in *Drosophila melanogaster*, *microRNA-7*, transcriptional factor *Yan*, and their downstream targets together form a feedback system to stabilize photoreceptor determination processes against temperature fluctuations[6]. These examples illustrate that a network perspective on how molecules function together in biological processes provides us with valuable insight into underlying molecular mechanisms.

Here, we explore the role of metabolic network properties in the context of biological aging. Aging arises from complex interactions among multiple biochemical and metabolic products[7–9], and these networks may decline in integrity with advancing age[10]. In the aged tissues of mice, for example, expression variation in single genes is significantly increased, and expression correlation between genes is decreased, relative to similar measures from young animals[11,12]. A causal link between longevity and network integrity has been suggested by data from fruit flies and nematode worms showing that specific anti-aging interventions, such as dietary restriction, promote the connectivity of transcriptomes and metabolomes[13,14]. Mechanistically, aging may diminish network integrity by preferentially affecting the "hubs", because molecules that have more partners[15–17] or that connect to multiple functional groups[18] are more likely to influence lifespan. Interestingly, metabolic network structures appear to be dynamically regulated—hub positions shift in response to environmental conditions (e.g., nutrient availability[19]), although the mechanisms responsible are largely unknown. Sensory perception is likely a key initiator of such changes, while the consequences may be substantial for integrative traits, such as health and lifespan.

Neuroendocrine systems are important mediators linking sensory perception, lifespan, and metabolic health[20,21]. In humans, serotonin signaling coordinates a range of behavioral and physiological traits, including emotion, sleep, feeding, and metabolism (e.g., refs., [22,23]). These regulatory mechanisms are well conserved across taxa, with many new insights first described in invertebrate systems, such as *Caenorhabditis elegans*[24–26] and *Drosophila*[27–29]. For instance, research from *C. elegans* revealed that, despite their well-known role in modulating appetite and feeding behavior[22,30], specific serotonin receptors regulate fat metabolism independent of feeding[25]. More recently, we discovered that the serotonin 2A (5-HT2A) receptor in *D. melanogaster* modulates lipid storage, stress resistance, and longevity in response to nutrient availability instead of nutrient intake[31].

While investigating how 5-HT2A is involved in coordinating dietary conditions with physiology and aging, we found that it was required for diet-dependent changes in the abundance of metabolites from central metabolic processes[31]. These included TCA cycle intermediates and their amino acid precursors, which govern energy homeostasis and nutrient flux through many metabolic networks. We therefore speculated that changes in these hub metabolites may affect the hidden structure of broader metabolic networks, and that such changes in network structures per se may modulate lifespan[10,13]. Indeed, we found that diet-dependent changes in many characteristics of network integrity, including measures of connectivity, average shortest distance, module clustering, and robustness are mediated by serotonin signaling through the 5-HT2A receptor. Our findings therefore ascribe a role for this ancient signaling pathway in regulating systems-level properties and, to our knowledge, provide the first demonstration of a molecular mechanism responsible for changes in metabolic network structure that may modulate aging.

## Results

**Dietary choice decreases metabolite co-expression through 5-HT2A.** Previously, we demonstrated that 5-HT2A modulates aging in *Drosophila* in response to how macronutrients are presented to the animals[31,32]. We found that dietary choice (i.e., presenting 10% w/v sugar and 10% w/v yeast in separate wells) increased mortality (Fig. 1a) and reduced mean lifespan of wild-type males when compared with siblings that were presented with a single, complete food of the same macronutrient composition (i.e., 10% w/v sugar and 10% w/v yeast mixture)[31]. A dietary switch at 20 days of adult life rapidly increased/decreased mortality when flies were transferred to a choice or fixed diet, respectively (Fig. 1b, c). Reduced expression of *5-HT2A*, caused by a transposable element insertion in its promoter region[33], eliminated the differences in mortality between the two dietary environments (Fig. 1d). 5-HT2A was also required for the effects of nutrient presentation on the metabolome: loss of wild-type *5-HT2A* gene function diminished the changes in metabolite abundances caused by dietary choice, where critical TCA intermediates and their amino acid precursors were significantly increased in wild-type animals when they had to choose between sugar and yeast before each meal[31]. The questions that we address herein are whether dietary choice, and serotonin signaling through this receptor, also affect metabolic network structures and whether these system-level changes influence health and aging.

Targeted metabolomics data were collected from the heads and bodies of 22-day-old adult flies that had been provided either a choice diet (CD) or a fixed diet (FD) starting at 2 days post eclosion. We also examined flies that were maintained on these diets until 20 days of age and then switched 2 days prior to sampling, either from a choice diet to one that is fixed (CD → FD) or vice versa (FD → CD). The age at which flies were switched was chosen as that time when the mortality differences between the choice and fixed diet cohorts were maximized (Fig. 1a). We began our interrogation of diet-dependent metabolic networks by calculating correlations between all pairs of metabolites across replicate samples within each diet treatment. We chose this approach because changes in physical networks (e.g., gene regulatory and metabolic networks) are often reflected in changes in the correlations between pairs of transcripts or metabolites[13], and previous studies have suggested that correlation coefficients ($\rho$) between molecules might decrease with age in fruit flies and mice[12,13].

We found that the correlations between metabolites in flies fed on a choice diet were significantly weaker than those from flies fed a fixed diet (Fig. 1e). We focused on the extreme tails of the correlation distribution. Visual inspection of the distribution suggested that while the mean across all correlations was not different, the tails of the distribution were considerably larger for fixed diet versus choice diet flies. The percentage of highly correlated metabolite pairs (i.e., those with $|\rho| \geq 0.8$) in the heads of flies fed the choice diet was 7.7%, compared to 13.9% from flies fed the fixed diet ($P < 0.001$, Fisher's exact test). A similar trend was observed in bodies (Fig. 1i), with a 28.2% reduction of highly correlated pairs in flies given a dietary choice (12.2% vs 17.0%, flies on choice vs fixed diet, respectively; $P < 0.001$, Fisher's exact test). The differences were reduced upon dietary switch (Fig. 1f, g, j, k). This partial reversal suggests a 48-h exposure to a new diet is

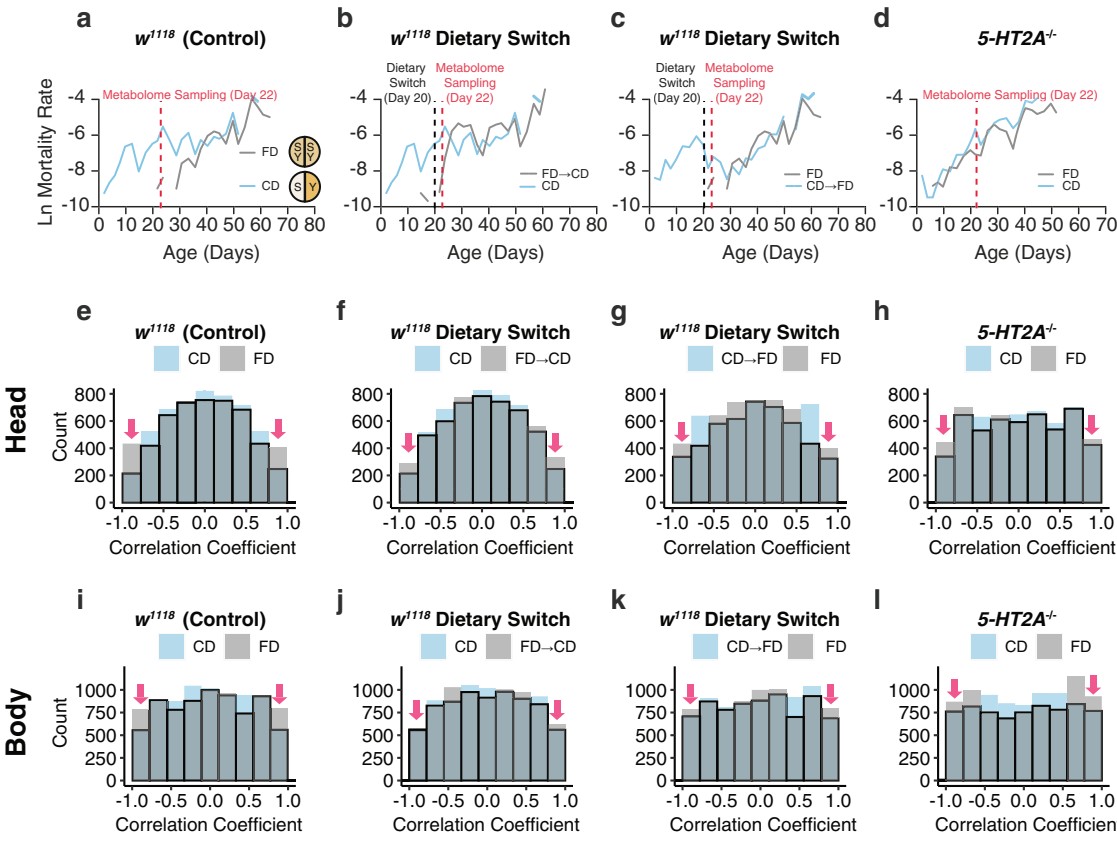

**Fig. 1 Dietary choice influences both mortality and metabolite correlations in metabolomes through serotonin signaling. a–d** Mortality data, which are from our recent publication (Lyu et al.[31] also see our permission statement), are used to demonstrate the association between aging and metabolite correlations under nutrient choice. Mortality rate is plotted on the natural log scale ($N = 225–282$). **a** Control males fed on a choice diet exhibit higher mortality rates early on compared to their siblings on a fixed diet. **b, c** Switching to a different diet (applied at 20 days of adulthood, as denoted by the dashed line) immediately changes the mortality rates that reflect the influences from the new diet. **d** 5-HT2A mutants are immune to the choice-induced mortality effects. **e–l** Dietary choice influences correlations between metabolites in metabolomes. Control males fed a choice diet have less highly correlated metabolite pairs (indicated by red arrows) compared to their siblings that fed a fixed diet in both heads (**e**) and bodies (**i**). Either dietary switch (**f, g, j, k**) or a mutation in 5-HT2A (**h, l**) diminishes the dietary differences, mirroring the effects on mortality.

sufficient to influence metabolite correlations, yet a complete transition (if possible) may take longer period of time as indicated by the mortality curves (Fig. 1b, c). These data reveal an effect on metabolite correlation structure in control animals that is associated with dietary choice and that is temporally coincident with changes in age-specific mortality.

Disrupting serotonin receptor 2A abrogated the differences in mortality between diets (Fig. 1d), which led us to ask whether the effect of diet on metabolite correlation structure was also mediated through the 5-HT2A receptor. Indeed, we found that the loss of highly correlated metabolite pairs in the choice diet environment was significantly reduced in mutant animals by roughly 39% in the heads (from a 6.2% decrease in control animals to 3.8% in 5-HT2A mutants, $P < 0.001$, Fisher's exact test) and 19% in the bodies (from a 4.8% decrease by choice in the control genotype to 3.9% in mutants, $P = 0.030$, Fisher's exact test). Our observations suggest that the way in which nutrients are presented influences the physical nature of the metabolome and that these changes in network structure are mediated, at least in part, by serotonin signaling.

To better understand the extent of changes in network structure induced by our dietary manipulation, we used adjacency matrices to construct correlation networks among metabolites. We considered two metabolites to be directly linked if they were significantly correlated with each other (significance was estimated using Spearman's rank-order correlation followed by

FDR correction). These connections formed the edges of our metabolite correlation networks that link individual metabolites or nodes (see Fig. 2a and "Methods" for more details on network construction). Biologically speaking, if two metabolites are connected by an edge, then they are thought either to interact directly (e.g., by co-involvement in the same biochemical reaction) or to share patterns of abundance, perhaps as a result of co-regulation by common mechanisms. Using this approach, we constructed one network for each combination of diet, genotype, and tissue sample. We used a threshold FDR = 0.10 to identify significant comparisons among different conditions, which is equivalent to $\rho = 0.7–0.8$ across different conditions (Supplementary Fig. 1). A more stringent criteria of FDR = 0.05 ($\rho = 0.8–0.9$ across different conditions, Supplementary Fig. 1) yielded similar results, but led to a restrictive number of edges in the head network derived from control flies on a choice diet (Supplementary Fig. 2). We found that networks constructed from each of the 12 groups exhibited a similar organization across diets, tissues, and genotypes (see Fig. 2b for networks constructed from heads and Supplementary Fig. 2 from bodies). Each consisted of one large, highly connected core group comprised of at least 70% of all the observed metabolites, together with many small satellite groups, each consisting of no more than five interconnected nodes, which are not connected with the core. Most of these satellites were comprised of a single, unconnected metabolite.

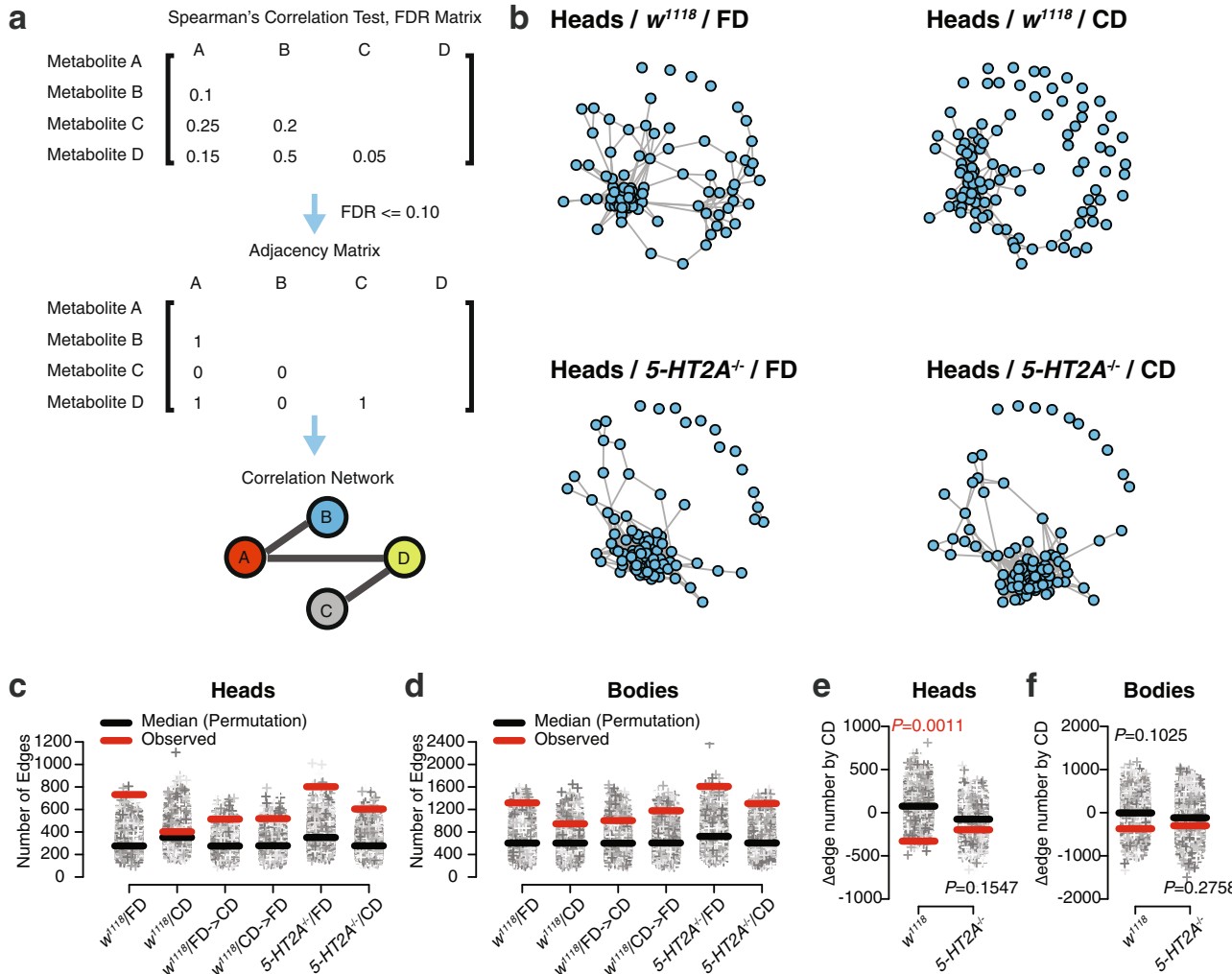

**Fig. 2 Serotonin signaling modulates choice-induced edge number reduction. a** Diagram illustrating correlation network construction. In brief, we estimated the significance of metabolites correlation using Spearman's rank-order correlation coefficient test. Then, adjusted $P$ values (by Benjamini–Hochberg procedure) were used for constructing adjacency matrices (with FDR = 0.1 as the cutoff). We used adjacency matrices to infer metabolite correlation networks, in which nodes represent individual metabolites and edges indicate links between them. **b** Visualization of the correlation networks using the Fruchterman–Reingold algorithm demonstrates similar organizations across different diets and genotypes. Each network consists of one large, highly connected core group together with many small satellite groups, which are disconnected from the core. Most of these satellites were comprised of a single, unconnected metabolite. **c**–**f** Permutation analysis is used to demonstrate whether the edge number differences between groups are significant. **c**, **d** Plots showing the edge number of real networks (denoted as the red bar) against randomized networks (gray dots represent edge number from 10,000 simulations, while black bars indicate the median of each group). In both heads (**c**) and bodies (**d**), the observed edge number is higher than the median of permutated ones from the same group, while the deviation is most extreme in flies that fed on a fixed diet, suggesting diet-dependent effects on edge number. **e**, **f** Edge differences between dietary conditions are further calculated to evaluate statistical significance. Permutated data are used as sampling distributions. Choice-induced edge number decrease is significant in the heads (**e**) of control males ($P = 0.0011$, permutation test). Such trend is also observed in bodies (**f**), yet the $P$ value is not significant ($P = 0.1025$, permutation test). Dietary differences are not seen in 5-HT2A mutants (**e**, **f**), suggesting these effects require serotonin signaling.

To understand the composition of the core group, we queried it for metabolic pathway enrichment. A complete enrichment analysis was constrained by the size of our metabolomic panel, so we instead chose to highlight pathways with more than six metabolites (as shown in Table 1). We found that three major metabolic pathways, including aminoacyl-tRNA biosynthesis, purine metabolism, and glycine, serine and threonine metabolism, were highly represented in all core groups. Four pathways (alanine, aspartate and glutamate metabolism, butanoate metabolism, arginine biosynthesis, and glyoxylate and dicarboxylate metabolism) that appeared in the controls fed on a fixed diet were not highly represented in the heads of the same genotype that fed on a choice diet. Two pathways (butanoate metabolism as well as

alanine, aspartate and glutamate metabolism), however, were highly represented in the heads of 5-HT2A mutant flies fed a choice diet.

**Serotonin signaling modulates edge number, connectivity, and average shortest distance of metabolite correlation networks.** We next asked whether the finer structures of the networks were influenced by dietary presentation. We began by focusing on edge number because of our results that revealed an influence of diet on correlation coefficients (Fig. 1e–l). Permutation analysis was used to compare the edge number of real networks with that of randomly shuffled ones ($n = 10,000$, see "Methods" for details) and to identify significant edge differences among dietary

**Table 1 Pathway analysis on the metabolites from each core group.**

| | # of metabolites in the pathway (heads) | | | | # of metabolites in the pathway (bodies) | | | |
| --- | --- | --- | --- | --- | --- | --- | --- | --- |
| | $w^{1118}$ | | 5-HT2A$^{-/-}$ | | $w^{1118}$ | | 5-HT2A$^{-/-}$ | |
| | FD | CD | FD | CD | FD | CD | FD | CD |
| Aminoacyl-tRNA biosynthesis | 20 | 16 | 18 | 17 | 19 | 19 | 19 | 18 |
| Purine metabolism | 11 | 11 | 11 | 13 | 12 | 12 | 13 | 13 |
| Alanine, aspartate, and glutamate metabolism | 10 | | 10 | 10 | 11 | 11 | 11 | 10 |
| Glycine, serine, and threonine metabolism | 9 | 7 | 9 | 9 | 9 | 9 | 9 | 9 |
| Butanoate metabolism | 7 | | | 7 | 7 | 7 | 7 | |
| Arginine biosynthesis | 7 | | | | 8 | 8 | 7 | 8 |
| Glyoxylate and dicarboxylate metabolism | 7 | | 7 | | 7 | 7 | 7 | 7 |

We demonstrate the metabolic pathways with the number of detected metabolites greater than six.

conditions. In the observed network derived from the heads of $w^{1118}$ flies fed a fixed diet, the number of edges placed the network in the upper extreme of the distribution of edges drawn from randomized networks ($P < 0.002$; red bar vs gray dots in the first column of Fig. 2c), while for the observed network drawn from heads of flies maintained on a choice diet, the number of edges was not unusually high ($P < 0.341$, shown in the second column of Fig. 2c). Differences of this nature were also observed in the networks constructed from body samples from the same cohorts: the observed edge number in the fixed diet network was more extreme ($P < 0.002$, as shown in the first column of Fig. 2d) than was edge number from the choice diet group ($P < 0.077$, the second column of Fig. 2d). To determine whether these differences were more than expected by chance, we compared the differences of edge number (#edge of CD − #edge of FD) between 10,000 pairs of randomly shuffled networks within each genotype and tissue. We found that the reduction in the number of edges induced by dietary choice was significant in the network constructed from the heads of control animals ($P = 0.0011$, permutation test, the first column in Fig. 2e). A similar trend was observed in the metabolic networks constructed from fly bodies, but the significance of diet was marginal ($P = 0.1025$, permutation test, the first column in Fig. 2f). If reduced edge number is associated with increased mortality and faster aging, then we would expect that manipulations that modulate mortality should also affect edge number. As also indicated by the correlation distributions, dietary switches, which led to rapid changes in mortality, also stimulated a 48-h modification of metabolic network edge number in the direction that reflected the influence of the new diet. The observed edge numbers of switched groups (depicted as the third and fourth data groups in Fig. 2c, d), are intermediate to that of the unswitched FD and CD groups (depicted as the first and second data groups in Figs. 2c, d), supporting the notion of an ongoing transition in metabolic network structures 48 h after dietary switch. These changes were observed in both heads and bodies.

We investigated whether receptor 5-HT2A was required for the reduction in edge number induced by dietary choice. We found that the diet effect was reduced in 5-HT2A mutant flies (Figs. 2c and 2d) compared to control animals, primarily due to more edges in the network constructed from the heads of flies that fed on a choice diet (Supplementary Fig. 3, permutation test,

$P = 0.0022$,). When fed on a fixed diet, genotype differences were not significant (Supplementary Fig. 3, permutation test, $P = 0.30$ and $P = 0.40$ in heads and bodies, respectively). In mutant flies, edge differences between diets were not statistically significant ($P = 0.15$ in heads and $P = 0.28$ in bodies, Fig. 2e, f), indicating that these effects are 5-HT2A dependent.

The observed changes in edge number indicated a dietary influence on network connectivity. This led us to examine the distribution of the number of edges connected to each metabolite, which is referred to as the degree, in our correlation networks. Metabolite degree exhibited a bimodal distribution under most conditions in both heads (Fig. 3a) and bodies (Supplementary Figure 4), where low-degree and high-degree nodes were clearly overrepresented. Notably, we observed a strong reduction in high-degree nodes in the heads of control flies fed a choice diet, but no effect of diet on degree distribution in 5-HT2A mutant (Fig. 3a). We defined high-degree nodes relative to core size (i.e., those with degree ≥30% × total node number), to exclude the possibility of confounding between the two (Fig. 2b). High-degree nodes accounted for 32.6–59.0% of the total metabolites in all conditions (indicated as red dots in Fig. 3a and Supplementary Fig. 3), except for measures from the heads of control flies fed a choice diet, where none of the nodes met the criteria. Interestingly, switching to a fixed diet for 48 h was sufficient to restore high-degree nodes, while switching to a choice diet for the same amount of time was not able to eliminate those nodes (Fig. 3a, second row). The loss of high-degree nodes was not observed in the bodies of the same cohort (Supplementary Figure 4), nor in the heads of 5-HT2A mutants (40.5% and 50.5% of the metabolites were accounted as high degree, respectively). This suggested that serotonin signaling reduced neural network connectivity by removing "hub" genes in response to dietary choice[34–36].

Because random pairs of metabolites are most likely to be connected through network hubs, we conjectured that the average path length between two metabolites in the network would also be affected by nutrient presentation. If true, we wanted to determine whether this effect was mediated by 5-HT2A. We measured the average of the shortest path length (see "Methods" for definition) across all metabolite pairs[37]. Generally, having fewer edges in the network results in fewer possible routes between two metabolites, thus increasing the distance between them, but this is not necessarily so. We found that the average shortest distance ($\bar{d}$) between metabolites was significantly increased in networks calculated from the heads and bodies of control flies in a dietary choice environment compared to the corresponding samples from the fixed diet (Fig. 3b, c, one-way ANOVA, $P(\text{diet}) < 0.001$ for control flies). These changes were dependent on 5-HT2A in both tissues (two-way ANOVA, $P(\text{diet}: \text{genotype}) < 0.001$). Interestingly, the values of $\bar{d}$ in these networks (ranging from 1.9 to 4.4) were small relative to the size of the networks (network diameters, which are the longest paths in the networks, range from 5 to 13). The relationship is consistent with small-world architecture[38,39], documented in many previously examined metabolic networks[34–36].

**Dietary choice induces the fragmentation of the network in a 5-HT2A dependent manner.** Biological networks are often organized into tightly knit modules, within which metabolite–metabolite interactions are dense and between which there are only looser connections[40]. Here, in metabolite co-expression networks, modules might represent a group of functionally related metabolites that respond to dietary inputs in a highly coordinated manner. We investigated dietary influence on the organizational structure of metabolites by comparing the modules from animals that fed on

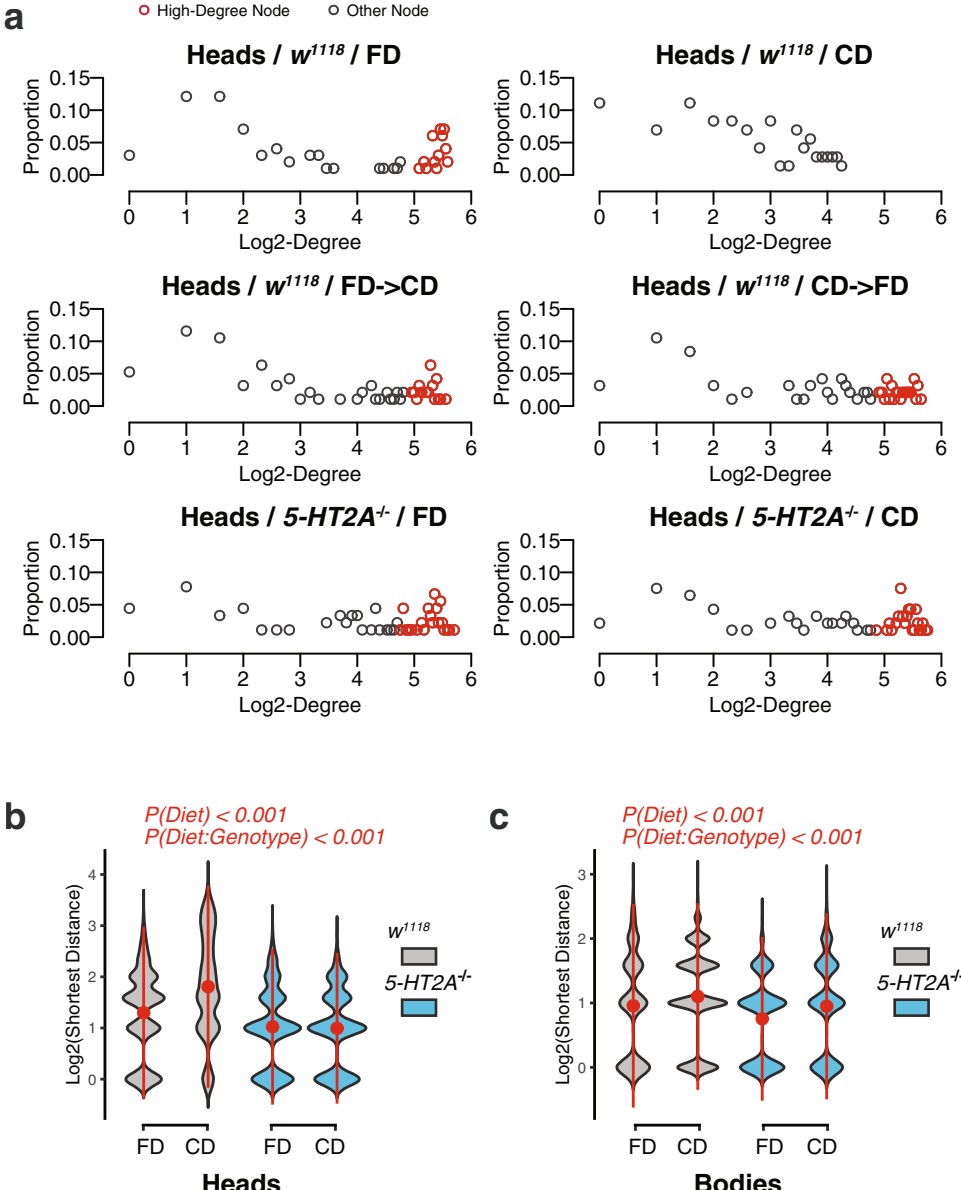

**Fig. 3 Choice environment reduces network connectivity and increases average shortest distance in a 5-HT2A-dependent manner. a** Dot plots showing the frequencies of node connectivity (degree) exhibit a decrease in the number of high-degree nodes in the heads of control flies that fed on a choice diet (compared to that of flies fed a fixed diet). Switching to a fixed diet for 48 h is sufficient to restore the high-degree nodes. Importantly, node connectivity of *5-HT2A* mutants is not influenced much by diet, suggesting that the dietary effects are mediated by serotonin signaling. **b**, **c** Exposure to a choice diet also increases the average of shortest distance in networks and such effects are dependent on 5-HT2A in both heads (**c**) and bodies (**d**). *P* values on the top of the violin plots are obtained from two-way ANOVA.

different diets. Metabolite modules were determined using Newman's leading eigenvector algorithm[41], which is built around the concept that the optimal metabolite modules are those with maximal "modularity score" (see "Methods" for details). We then visualized the modules by plotting the correlation between every pair of metabolites. Once metabolites were ranked by the group order, modules were represented as visually perceivable blocks (e.g., Figs. 4a, b).

We found that dietary choice induced network fragmentation in heads (Fig. 4a). A greater number of smaller modules were identified in the networks constructed from the heads of flies given a choice diet compared to those given a fixed diet. Dietary effects in module numbers were partially reversed by switching to another diet. Switching to a choice diet for 48 h increased the module number (from 7 to 11, Supplementary Fig. 5a), while

switching to a fixed diet for the same amount of time dramatically decreased the module number (from 31 to 12, Supplementary Fig. 5b). The two dominant modules from flies given a dietary choice were comprised of only 21 metabolites each, while the corresponding modules from fixed diet networks contained 47 and 44 metabolites, respectively. Dietary differences were not observed in bodies (Supplementary Fig. 6). We speculated that the highly interconnected dominant module (indicated by the yellow bar in Fig. 3a, fixed diet) represented a homeostatic central unit, wherein metabolites respond to environmental input in a coordinated fashion. We examined the functional enrichment of this unit using MetaboAnalystR[42]. Using our threshold of $N > 6$ metabolites (only 10–27% of the total functional categories fit this criterion), we found that purine metabolism ($N = 9$) and aminoacyl-tRNA biosynthesis ($N = 8$) were highly represented

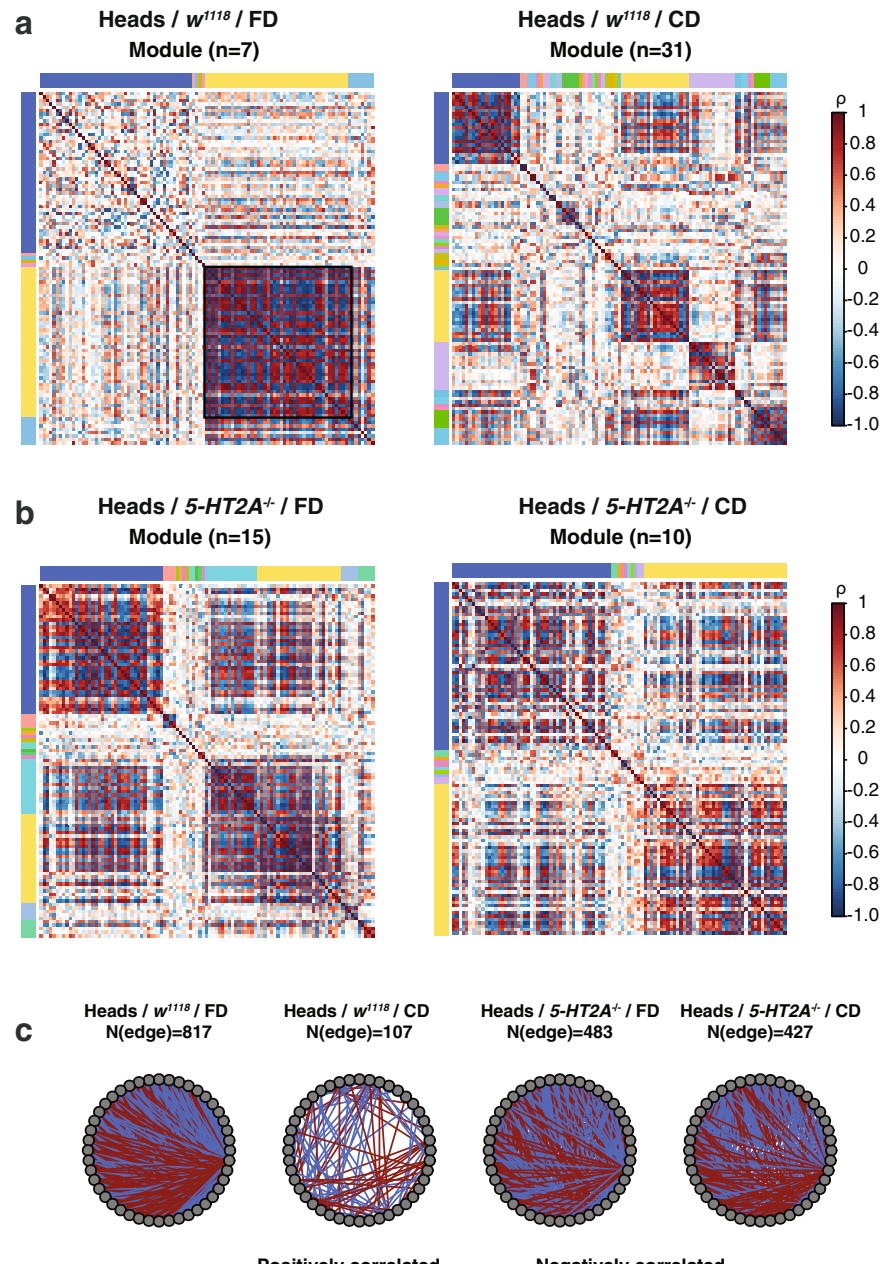

**Fig. 4 Dietary choice induces the fragmentation of correlation networks partially through 5-HT2A. a, b** Correlation plots showing the dietary influences on the community structures in both control males (**a**) and *5-HT2A* mutants (**b**). Color bars on the side and the top of the correlation plots indicate individual modules, with yellow and blue blocks representing the dominant ones (those who have the most metabolites out of all the modules) in each group. **a** In the head metabolomes of control flies that fed on a choice diet, we observed an increase in the number of modules and a decrease in the size of the dominant modules, compared to that of their siblings from a fixed diet. **b** Dietary differences are eliminated in *5-HT2A* mutants. **c** Metabolite–metabolite interactions in the yellow major group are largely lost in the CD networks that are derived from the heads of control males, while part of them are restored in the *5-HT2A* mutants.

in the dominant module in fly heads. Our results suggest that central metabolic processes are less coordinated in a heterogeneous nutritional environment, in which animals must choose their meal than they are in a more homogenous one.

We next asked whether the observed network fragmentation that was induced by our dietary manipulation required serotonin signaling through the 5-HT2A receptor. Correlation plots revealed that dietary influences on the number and size of modules were largely abrogated in the heads of *5-HT2A* mutants (Fig. 4b), as these flies exhibited an decreased/increased module number when fed on a choice or a fixed diet, respectively. Notice for mutants

that fed a fixed diet, despite their network connectivity remained the same as control flies on this diet (Fig. 3a and Supplementary Fig. 3), their community structures showed differences. Such changes are associated with the increased mortality of *5-HT2A* mutants on a fixed diet (Fig. 1a, d), suggesting network fragmentation may have more implications in aging than overall connectivity. To further quantify the influence of 5-HT2A function on the dominant module, we calculated the number of metabolite interactions (i.e., network edges) that were lost in control animals, but maintained in mutant flies when both were placed on the choice diet. Interactions between metabolites inside

of the dominant module were reduced by 86.9% by dietary choice (Fig. 4c), while the effect of dietary choice on edge number was much smaller in networks from *5-HT2A* mutant flies (reduced by only 11.6%, $P < 0.001$, Fisher's exact test), which suggests that 5-HT2A is partially required for reduced modularity following dietary choice. Of the 717 connections that were lost in the heads of control flies when they were switched to a choice diet (see Fig. 4c), 317 were restored in *5-HT2A* mutants. For those that were restored, 79.8% (253/317) were restored in the same direction (either positive or negative), as originally observed in networks from flies fed a fixed diet. Together these data suggest that serotonin signaling affects both the size of the central metabolic unit, as well as the number of modules. Specifically, 5-HT2A signaling appears to mediate the loss of metabolite interactions in the central module that affects amino acid metabolism and energy homeostasis upon dietary choice, which may be influential on mortality (Fig. 1a–d).

**Dietary choice decreases network robustness through serotonin signaling**. A less connected and more fragmented network might be considered less robust and less resilient to perturbation. We examined this issue computationally, using two approaches to study the effect of node removal on the integrity of the network core. First, we measured the effect of random node removal on the average shortest distance ($\bar{d}$) of each network core, as $\bar{d}$ reflects how well metabolites are connected with each other in networks. It has been noted that biological networks are generally robust to random errors[43], and indeed, we found that removal of up to 30% of the nodes (e.g., 20 nodes from the networks constructed from head and 30 nodes from body samples) had no significant effect on $\bar{d}$ in most conditions. (Fig. 5a, b). The only influence is in the head of control flies that fed on a choice diet, where node removal decreased $\bar{d}$ (indicated as black triangles in Fig. 5a). Our second computational approach for studying robustness involved targeted network "attack"[43,44], in which node removal was prioritized based on measures of node importance, or "centrality". We used two measures of node centrality, degree and eigenvector, which yielded similar results (see "Methods"). First, we observed a consistent increase in $\bar{d}$, when nodes with the highest centrality were removed first and then subsequent nodes were removed in descending order (Fig. 5c, d for degree centrality; Fig. 5e, f for eigenvector). Second, attacking nodes in the metabolomic networks measured from the heads of flies fed a choice diet resulted in several rises in $\bar{d}$, with the removing order based on either degree (Fig. 5c, d) or eigenvector (Fig. 5e, f). Two metabolites, 5-aminopentanoic acid and choline, were influential in both attacking methods, while GMP, isovalerate, carnitine, and glutarate were specific to one method (Fig. 5c, e). Our data suggest that dietary choice may act through serotonin signaling to reshape the connections between metabolites in such a way as to produce a network that is relatively more vulnerable to node removal.

**Dietary choice increases phenotypic vulnerability to genetic perturbations on the glutamine–αKG axis**. Our systems analyses motivated us to propose a working mechanistic model (Fig. 6a), in which the effects of dietary choice on network structure, which include increased fragmentation and average shortest distance, as well as reduced connectivity and robustness, would increase organism-level vulnerability to genetic or environmental perturbations, akin to removal of network nodes, as measured by resistance to stress. Previously, we demonstrated that the glutamine–α-ketoglutarate (αKG) axis likely acts downstream of serotonin signaling to mediate the lifespan effects from dietary choice[31]. As a key TCA intermediate and a

cofactor of DNA/histone demethylating enzymes, αKG sits at the intersection of mitochondrial metabolism and epigenetic regulation of aging[36]. Manipulating this hub metabolite presumably influences multiple physiological pathways and might affect systemic robustness.

We therefore executed adult-specific, RNAi-mediated knockdown of each of three key enzymes that control the glutamine–αKG axis: *GDH* (*glutamate dehydrogenase*), *GS1* (*glutamine synthetase 1*), and *GS2* (*glutamine synthetase 2*; Fig. 6b), and we measured the effect of diet on starvation resistance (see "Methods" for details). We predicted that the effect of knockdown on this phenotype would be magnified in flies fed a choice diet because of the putative fragility of the observed networks (Fig. 6a). Consistent with our previous findings[31], control males that were fed a choice diet for 10 days exhibited a smaller decline in starvation resistance relative to their siblings fed a fixed diet (Fig. 6c and Supplementary Data 1). Importantly, we observed a consistent trend that knocking down any one of the three enzymes exacerbated the differences between a choice and a fixed diet, as predicted by our model (Fig. 6d–f and Supplementary Data 1; but note that the $P$ value associated with *GS1* knockdown does not reach significance, $P$(interaction) = 0.13). This effect was not caused by the transcriptional inducer, RU486, because control animals (*tub-GS-GAL4* coupled with *w*) responded as expected to dietary choice when RU486 was presented (Fig. 6c). While our data are consistent with the predicted model (Fig. 6a), whether such influences are due to changes in network structures or in individual metabolites (e.g., αKG, as indicated by our previous study[31]) remains to be determined.

## Discussion

We discovered that nutrient presentation modulated fundamental network structures in metabolomes through serotonin signaling via the 2A receptor branch. We probed different aspects of network integrity, including: (i) connectivity and average shortest distance, which revealed the degree to which the abundance of individual metabolites was responsive to the changes in the rest of the metabolomes; (ii) community structure, which reflected the underlying organizations of the correlational networks, and (iii) robustness, which predicted how well the systems could handle extrinsic and intrinsic perturbations. Our analyses revealed that meal choice reshaped a highly interconnected metabolic network into a more fragmented one, which consisted of fewer edges and smaller communities, and that these structures were more vulnerable to perturbations. Importantly, all these effects were dependent to some extent on receptor 5-HT2A. While similar effects on metabolomes have been observed following dietary restriction[13], our findings provide a molecular mechanism between nutrient sensing and network integrity, which, to our knowledge, are the first to do so.

We speculated that the changes on network structures through 5-HT2A were adaptive responses to a complex nutritional environment. Serotonin signaling has been shown to influence both energy states and behavior[25,30], and it seems plausible that 5-HT2A signaling is involved in coordinating a variety of responses to environmental nutrients in such a way as to maximize individual fitness. Although a less connected, more fragmented metabolome may be associated with a reduced lifespan[13], it may be beneficial to overall fitness in a heterogenous environment, in which specific metabolic pathways might be responsive to individual nutrients. Improved metabolic efficiency and specialization in somatic cells are traits that are thought to be favored over longevity assurance[45,46].

We propose that the elevated mortality rates that flies experience when given a dietary choice may be due, at least in part, to a

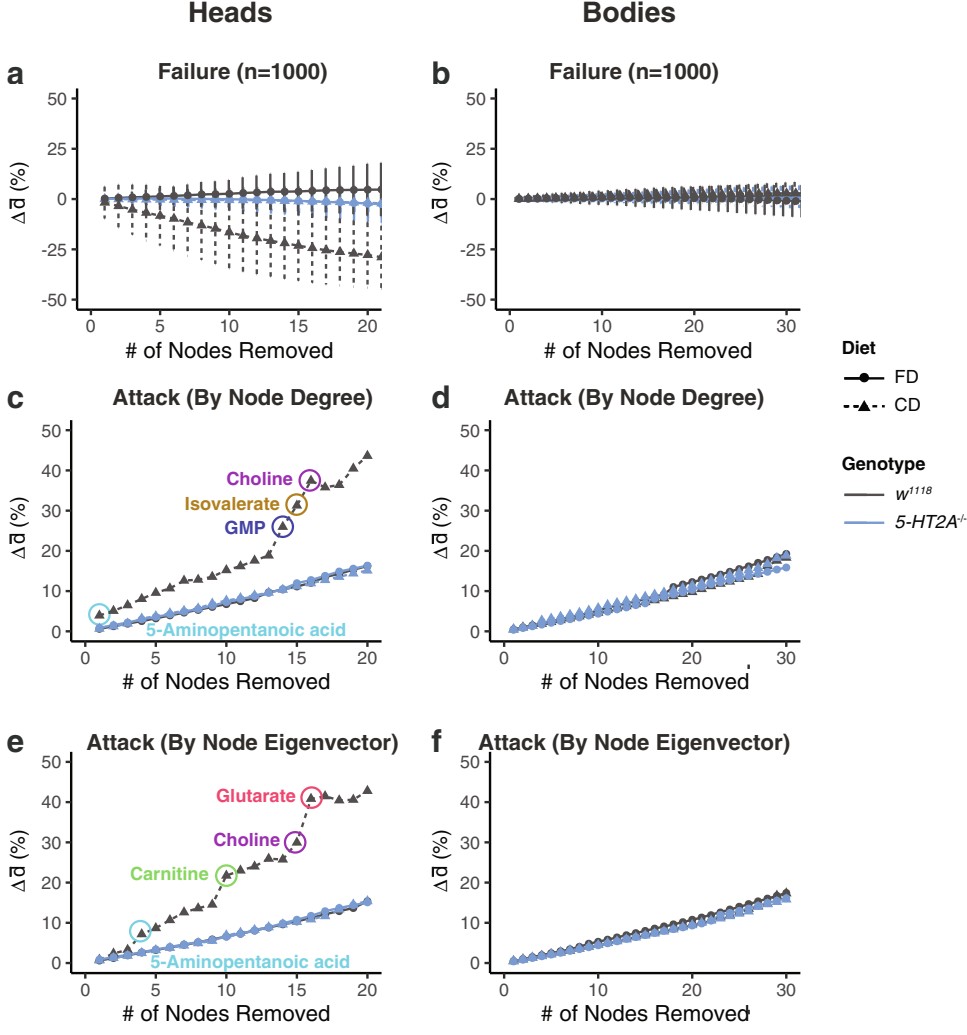

**Fig. 5 Serotonin signaling mediates the decrease in network robustness in response to nutrient choice. a, b** Randomly removing nodes in the networks does not increase average shortest distance in neither heads (**a**) nor bodies (**b**). Dots and error bars indicate the mean and standard deviation from 1000 simulations, respectively. **c–f** Sequential removals following node centrality (estimated from node degree (**c**, **d**) or node eigenvector (**e**, **f**)) increase the average shortest distance ($\bar{d}$) of correlation networks in both heads and bodies. Importantly, such increases are more significant in the heads of control flies that fed on a choice diet, suggesting this network is more vulnerable than other ones. Key metabolites that lead to the differences are highlighted in circles.

decrease in physiological robustness. A more fragmented, isolated network might be expected to increase organismal frailty and reduce resistance to other stressors, which was also indicated by our computational simulations (Fig. 5). Our experimental results showed that the effects of genetic perturbation on starvation resistance were magnified when flies were given a dietary choice, which is consistent with this idea (Fig. 6). This conjecture is also in line with previous findings that demonstrated a relationship between network integrity and aging in the transcriptomes and metabolomes of organisms across taxa, including nematodes[14], fruit flies[13,47], and mice[11,12,48]. The consistency of this trend across evolutionarily distant species suggests that a decline of systematic robustness is an emergent property of aging. Its characteristics may offer useful aging biomarkers, which have proven elusive at the molecular level[49].

Our study is among the few showing that network robustness and physiological robustness are associated, and that systems-level adaptations to dietary conditions can be mediated by single genes, providing new insight into the mechanisms through which network integrity impacts aging. Knocking down critical metabolic enzymes was more deleterious in animals with vulnerable

network structures (i.e., flies fed a choice diet, see Fig. 6d–f). On the other hand, loss of serotonin receptor 5-HT2A increased network robustness in a choice environment and also rescued choice-induced declines in starvation resistance[31]. Nevertheless, the preservation of network structure alone may not provide a complete picture of the mechanisms that influence physiological robustness, as the manifestation of biological robustness occurs on many different levels[3]. To better understand these systems, future experiments should attempt to isolate effects caused by network change from those caused by changes in individual molecules.

Going forward, a deeper understanding of the mechanisms through which serotonin signaling modulates lifespan may benefit from a focus on highly connected "hub" molecules that we have identified in the metabolic networks. The nature of biological networks concentrates significant power in these hubs because they often govern major fluxes of metabolic information[34,50], and network hubs are often effective targets for interventions that accelerate or slow aging[16–18,36]. At the molecular level, metabolic enzymes that sit at the branch points of metabolic reaction networks are considered more influential in

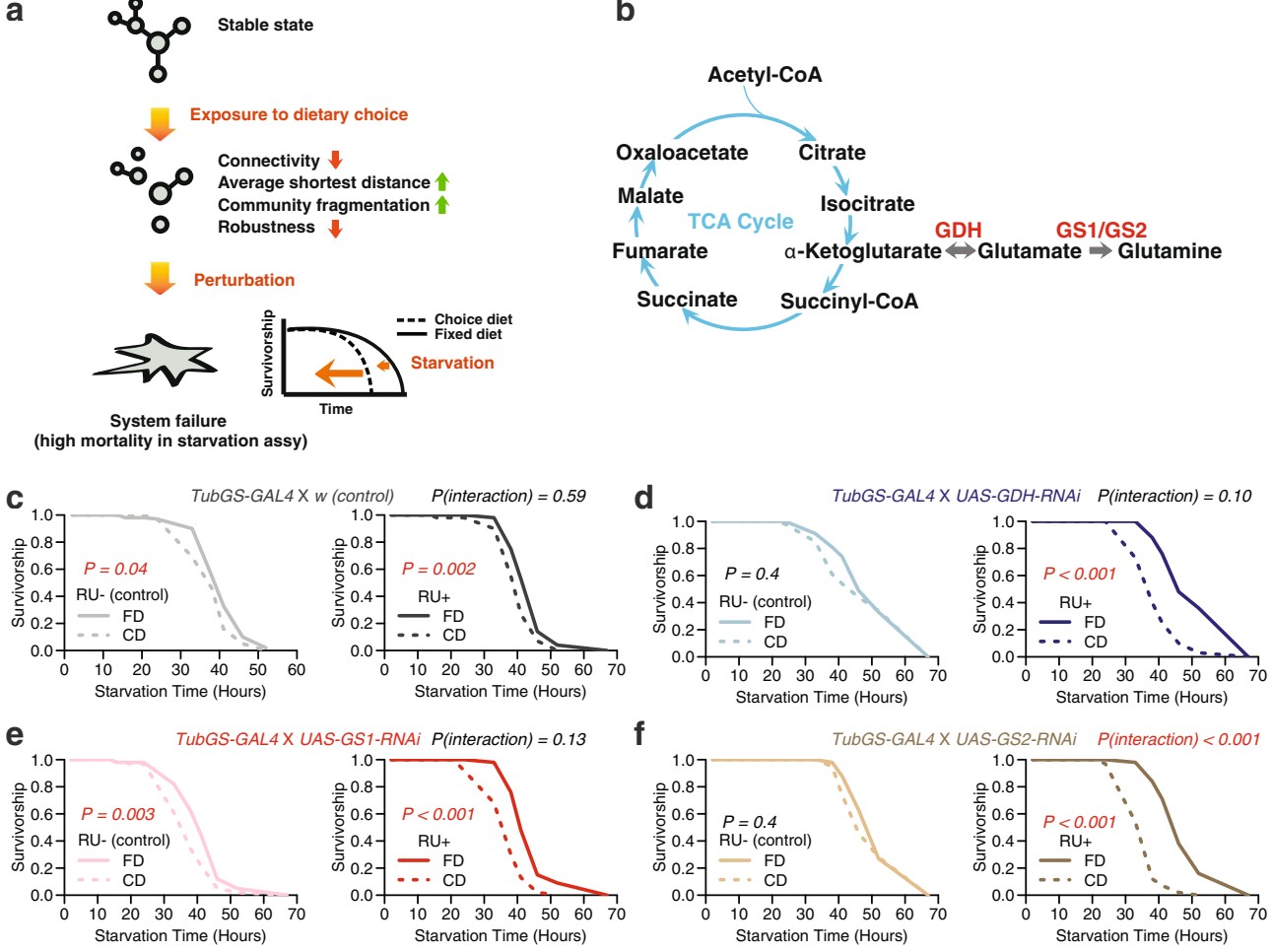

**Fig. 6 Exposure to dietary choice increases vulnerability to genetic perturbations. a** Our working model. **b** glutamine–αKG axis plays important roles in fueling the TCA cycle, one of the central biochemical reactions in metabolic networks. **c–f** Upon starvation, knocking down one of the key enzymes on the glutamine–αKG axis (*GDH*, *GS1*, and *GS2*) exacerbates the dietary effects on starvation resistance, suggesting flies that are preexposed to a choice diet are more sensitive to genetic perturbation. *P* values showing the dietary effects are obtained by log-rank test. We also use Cox-regression analysis to report *P* value for the interaction term between diet and genetic manipulation. Each treatment consists of six biological replicates (*N* = 5–10 flies for each vial).

these processes[51]. Recent studies have exposed the identities of hub metabolites and their roles in modulating lifespan. Metabolites considered as the downstream integrators of nutrient metabolism and signaling pathways, including nicotinamide adenine dinucleotide, reduced nicotinamide dinucleotide phosphate, αKG, and β-hydroxybutyrate, have emerged as key mediators of lifespan[36]. This is consistent with our observations that αKG appears to be one of the major effectors of serotonin signaling that modulates aging[31] and physiology (Fig. 6d–f). The αKG/glutamine synthesis pathways may also be important in modulating the ability of diet restriction to increase lifespan[19]. More recently, it has been shown that the lifespan-extending effects of αKG are preserved in both flies[52] and mice[53]. In the future, it is imperative to investigate the impact of these metabolites and metabolic pathways on network structures and metabolic communications, and to better understand the mechanisms through which nutrient presentation influences lifespan.

## Methods

**Metabolomic study.** Metabolomics data were collected in a separate study[31]. We briefly summarized our methods here, while the full details are included our publication[31]. Males were exposed to the choice or fixed diet for 20 days (food was changed every 2–3 days) and then switched to the other diet (experimental groups) or remained on the same diet (control groups). Forty to 50 heads or bodies were collected and homogenized for 20 s in 200 μl of a 1:4 (v:v) water:MeOH solvent mixture. Following the addition of 800 μl of methanol, the samples were incubated for 30 min on dry ice and homogenized again. The mixture was spun at 13,000 RPM for 5 min at 4 °C, and the soluble extract was collected into vials. This extract was then dried in a speedvac at 30 °C for ~3 h. Using a LC-QQQ-MS machine in the MRM mode, we targeted 205 metabolites in 25 metabolic pathways, in both positive MS and negative MS modes. After removing internal controls and any metabolites missing from >8 out of 76 head samples (10.5%) and 8 out of 78 body samples (10.3%), we were left with 103 head metabolites and 125 body metabolites. Metabolite in this data set were log-transformed and imputed using the *k*-nearest neighbor algorithm with the impute package in R Bioconductor (www.bioconductor.org). Afterward, we removed batch effects using the ComBat algorithm in the R package sva, then normalized the data to the standard normal distribution ($\mu = 0$, $\sigma^2 = 1$).

**Correlation network analyses.** Network analyses in this study were performed with the R language (version 4.0.5) within RStudio (version 1.2). Raw data and scripts are available as a GitHub repository (https://github.com/ylyu-fly/Metabolomics-FlyChoiceDiet).

*Correlation analysis and network construction.* To assess the relationship between metabolites for each biological condition, we estimated the significances of correlations using Spearman's rank-order correlation coefficient test, which was computed from the abundance of metabolites within biological conditions (*N* = 8–10). Afterward, *P* values were adjusted by FDR (Benjamini–Hochberg procedure[54]), which we used for constructing adjacency matrices A = {$a_{ij}$} and to infer the

correlation network (nondirected graphs, as no directions were involved):

$$a_{ij} = \begin{cases} 0, & FDR > 0.1 \\ 1, & FDR \le 0.1 \end{cases}$$

FDR = 0.1 was chosen as the cutoff followed standard practice. In correlation networks, nodes represented metabolites while edges represented links between them (where $a = 1$). Visualization via the R package igraph[55] revealed similar structures among the correlation networks across different diet and genotype groups. We observed a large, interconnected core group that represented the majority of the metabolites, which was accompanied by small groups as well as orphan metabolites. A pathway analysis using MetaboAnalystR[42] was further applied to show the composition of the core group.

*Permutation analysis.* We generated permutated datasets to estimate the probability of observing the edge number differences between diets. We used the existing values with the group identify (i.e., genotype and diet) being shuffled for each metabolite. $P$ values were obtained from comparing the real observation to 10,000 permutations.

*Measures of network integrity.* To investigate dietary influences on network integrity, we investigated four network attributes including

(1) Connectivity (i.e., edge number/node degree);
(2) Average shortest distance ($\bar{d}$);
(3) Community (module) structures, which reveal the underlying grouping pattern of correlation networks;
(4) Robustness, which is used to assess the influences of node removal on network attributes. We focused on the effects on average shortest distance ($\bar{d}$), as it represents metabolic communication responsiveness which might impact aging[56]. Here, we studied the robustness of network cores following our practice on the measures of average shortest distance.

*Connectivity.* We examined the dietary influences on network connectivity, specifically on the total edge numbers of the networks and on the number of neighbors for each node (i.e., degree). For total edge numbers, permutation testes were used to test whether the edge numbers in real networks were significantly deviated from those in random networks. Random networks were generated from 10,000 permutations; within each, group labels for each metabolite were shuffled independently. $P$ values were generated from the quantiles of the edge number (real experimental data versus the simulations). Similarly, $P$ values of the edge differences between diets (i.e., $\triangle_{edgenumber}$, defined as $N_{edge}(CD) - N_{edge}(FD)$ in each realization) were calculated for both control and mutated flies. In addition to examining the dietary influences on the total edge number of networks, we also investigated the dietary effects on the frequency of low-, intermediate-, and high-degree nodes by plotting the distribution of edge number per node.

*Average shortest distance.* Dietary effects on edge number suggested potential changes in how well metabolites connected to each other in these networks. To test this, we computed $\bar{d}$, the average shortest distance between any two metabolites (in the core structures where every node was linked to each other), with the R package igraph[55]. As core groups are connected networks, there exists at least one path between any two metabolites in the group (i.e., a sequence of adjacent metabolites $m_0, m_1, \ldots m_i$, from $m_0$ to $m_i$ each connected by at least one edge). The fewer steps along the shortest path from $m_0$ to $m_i$, the closer the two metabolites are considered to be.

*Module analysis.* We investigated the effects of meal choice on the community structures of correlation networks. Metabolites were separated into different modules using the "leading eigenvector" approach[41], with its implementation in igraph. The heart of this method is to obtain an optimized community classification by maximizing the modularity score (i.e., the number of edges within groups relative to that of random equivalent networks) across modules. Community structures were further visualized by the R package pheatmap.

*Network robustness analysis.* We assessed the dietary influences on network robustness, with a focus on the effects of node removals on the average shortest distance ($\bar{d}$). Following the practice of Albert et al.[43], two approaches were used here to perturb network cores, where we either randomly removed nodes (denoted as network failure), or targeted the nodes that were considered as the "hubs" of networks (denoted as network attack). We demonstrated the effects of network failure by showing the mean and standard deviation of $\Delta\bar{d}$ (changes in average shortest distance) from 1000 simulations. As for network attacks, sequential removal was executed based on node centrality, where nodes with the higher centrality being removed first. The effects of sequential removal (either through network failures or attacks) were demonstrated as the trajectory of $\Delta\bar{d}$.

### Genetic perturbation analysis

*Fly stock and husbandry.* The laboratory stocks $w^{1118}$, *UAS-GDH-RNAi* (BDSC#51473), *UAS-GS1-RNAi* (BDSC#40836), and *UAS-GS2-RNAi* (BDSC#40949)

were purchased from the Bloomington *Drosophila* Resource Center. *Tub-GS-GAL4* flies were generated in the Pletcher laboratory. All fly stocks were maintained on a standard cornmeal-based larval growth medium (produced by LabScientific Inc. and purchased from Fisher Scientific) and in a controlled environment (25 °C, 60% humidity) with a 12:12 h light:dark cycle. We controlled the developmental larval density by manually aliquoting 32 μl of collected eggs into individual bottles containing 25 ml of food. Following eclosion, mixed-sex flies were kept on SY10 (10% (w/v) sucrose and 10% (w/v) yeast) medium for 2–3 days until they were used for experiments. Pioneer table sugar (purchased from Gordon Food Service, MI) and MP Biomedicals™ Brewer's Yeast (purchased from Fisher Sci.) were used through our study.

*Dietary environments.* To study the effects of dietary choice, we created food wells that were divided in the middle (Fig. 1a). This allowed us to expose the flies to two separate sources of food simultaneously. For these experiments, we either loaded different foods on each side (choice diet) or the same food on both sides (fixed diet). In this study, choice diet contained 10% (w/v) sucrose on one side and 10% (w/v) yeast on the other, while fixed diet contained a mix of 10% (w/v) sucrose and 10% (w/v) yeast on both sides.

*Drug administration.* We knocked down metabolic enzymes using the GeneSwitch system[57]. The transcriptional inducer RU486 (mifepristone) was purchased from Sigma-Aldrich. Drug was first dissolved in 80% (v/v) ethanol at 10 mM concentration, marked by blue dye (5% (w/v) FD&C Blue No. 1; Spectrum Chemical), and stored at −20 °C. For experimental food, 200 μM RU486 or the same dilution of control vehicle (80% (v/v) ethanol) was made from the stock and added to the food.

*Starvation assay.* We used a starvation survival assay to test the effects of genetic perturbation on organismal robustness. Once-mated, 2–3-day-old male flies were kept on experimental (RU+) or control (RU−) food for 10 days (food was changed every 2–3 days) to activate the GeneSwitch system[57]. Afterward, we transferred flies to fresh vials containing 1% (w/v) agar. The number of dead flies was recorded approximately every 2–5 h using the DLife system[58]. Throughout the entire experiment flies were kept in constant temperature (25 °C) and humidity (60%) conditions with a 12:12 h light:dark cycle.

**Statistics and reproducibility.** The layout of metabolomic and physiological experiments is designed to exclude the influences from nonbiological factors (e.g., sample location). Sample size and the number of biological replicates are included in figure legends. Fisher's exact test was applied to analyze the effects of diet and genotype on the number of highly correlated metabolite pairs (defined as $|\rho| \ge 0.8$) against that of the rest pairs (i.e., $|\rho| < 0.8$). Permutation analyses were used to assess the dietary influences on the number of edges in the correlational networks. To test the effects of diet and genotype involved in average shortest distance ($\bar{d}$), we performed two-way ANOVA (on the log-transformed $\bar{d}$). In the starvation resistance assay, pairwise comparisons between different treatment survivorship curves were carried out using the statistical package R within DLife[58]. For testing the interaction between genotype and diet, we used Cox-regression analysis[59] to estimate $P$ value for the interaction term.

**Reporting summary.** Further information on research design is available in the Nature Research Reporting Summary linked to this article.

### Data availability
Metabolomic data are available from our GitHub repository (https://github.com/ylyu-fly/Metabolomics-FlyChoiceDiet)[60]. Source data for Fig. 6 are provided as Supplementary Data 1.

### Code availability
R scripts (R version 4.0.5) for metabolomics and network analyses are available from our GitHub repository (https://github.com/ylyu-fly/Metabolomics-FlyChoiceDiet)[60].

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

## Acknowledgements

We would like to thank the reviewers for constructive criticism that have resulted in a considerably improved manuscript, and Kelly Jin from the Promislow laboratory for her help with constructing correlational networks. This research was supported by the US National Institutes of Health, National Institute on Aging (R01 AG051649, R01 AG030593 to S.D.P. and R01 AG049494, R01 AG057330, and NSF grant DMS1561814 to D.E.L.P.), the Glenn Medical Foundation (to S.D.P.), and the Burroughs Wellcome Fund Collaborative Research Travel Grant (No. BWF1017452 to Y.L.).

## Author contributions

Y.L. conceived the idea, designed and performed the experiments, analyzed the data, and wrote the original draft. D.E.L.P and S.D.P supervised the study, provided financial support, and coordinated research activity planning and execution. All authors edited and critically revised the manuscript for important intellectual content and gave final approval for the version to be published.

## Competing interests

The authors declare no competing interests.
