## [Peer Review File · Communications Biology]

Reviewers' comments:

Reviewer #1 (Remarks to the Author):

The authors present evidence that diet-dependent changes in connectivity, average shortest distance, module clustering, and robustness are mediated by serotonin signaling through the 5-HT2A receptor.

The work is well structured and the conclusions are supported from the original observations. I find the work overall novel, relevant to the aging/endocrine field and I recommend that it is published as is with no further revisions.

Reviewer #2 (Remarks to the Author):

Summary

Lyu, Promislow, and Pletcher perform additional analyses and computational network modeling on previously published metabolomics data (Lyu et al. 2021. *Elife* 10, e59399) and find that measures of network integrity decrease with dietary choice in a 5-HT2A serotonin receptor dependent manner. In their previous work, the authors found that dietary choice – consuming protein and sugar components of the diet separately, rather than as a mixture – reduces lifespan and induces metabolic reprogramming in a 5-HT2A dependent manner. Metabolite changes induced by the choice diet included TCA cycle potentiation and amino acid metabolism; knockdown of glutamate dehydrogenase abrogated the effect of dietary choice on lifespan.

In this manuscript, rather than focusing on specific metabolites or pathways, the authors consider metabolic networks as a whole and hypothesize that metabolic network structure itself may modulate lifespan. In accordance with data from Lyu et al. *eLife* 2021, loss of 5-HT2A rescued network remodeling by the choice diet. The authors propose a model in which choice diet-induced metabolite network fragmentation would increase organismal vulnerability to perturbation. To test this model, they place flies on a choice diet to reduce network connectivity, add a “genetic perturbation” by knocking down enzymes that will affect alpha ketoglutarate levels, and finally assess starvation sensitivity. In accordance with their model, RNAi flies fed a choice diet had reduced starvation resistance compared with both WT flies and genetical control flies on a fixed diet. Experimentally linking disruption of metabolic networks to lifespan and mortality would be of great interest to many readers, but additional experimentation would be required to demonstrate this unequivocally.

Major comments

The network analysis was fascinating, as was the nearly complete rescue of the effects of choice diet on network structure by 5-HT2A knockout. The major hypothesis laid out in the introduction that metabolic network fragmentation itself may regulate lifespan is intriguing. I have two major critiques. First, the *eLife* 2021 paper noted interesting effects of dietary switching between fixed and choice diets on lifespan, which were not well addressed in this manuscript. More significantly, the hypothesis testing experiment in Fig. 6 was ultimately unsatisfying in addressing the major questions posed by the authors.

In Fig. 1, the effects of dietary switching on lifespan are noted and metabolite correlations are shown for FD>CD and CD>FD. The effect of dietary switching on metabolite correlation is notable in heads (1F, 1G). In FD>CD heads (1F), there's a small increase in metabolite correlation vs. choice diet – similar to the differences in 1E, the implications of which aren't discussed. Even more confusingly, in CD>FD heads, there seems to be an increase in the most extreme correlations on FD, but the second-to-extreme bin in the CD>FD flies. Is this really meaningful change at all? The only mention of these panels is in lines 130-131.

When it comes to network construction (Fig. 2 B, Fig. S1) these switching studies are left out

entirely, though edge analysis for these conditions is shown in Figs. 2C and 2D. Beginning on line 204, you state, "Indeed, dietary switches, which led to rapid changes in mortality, also stimulated a 48hr modification of metabolic network edge number in the direction that reflected the influence of the new diet. These changes were observed in both heads and bodies (depicted as the 3rd and 4th columns in Figure 2C and 2D), and they mirror the changes in mortality." However, particularly in heads, the FD>CD and CD>FD observed edges are nearly identical to one another. 48 hours is not a long time to remodel metabolic networks – these networks are likely still in flux toward their final state. Given the focus of the rest of the manuscript on serotonin receptor effects on network connectivity, the inclusion of these seemed like an underanalyzed afterthought that needs more consideration and interpretation.

The effects of food choice on network fragmentation is clearly presented with multiple analyses that were well described to this non-metabolomic-modeling reader! However, the broad hypothesis from the introduction that metabolic network fragmentation itself may regulate lifespan is not actually tested. I concur with the legend for Fig. 6, which states that, "Exposure to dietary choice increases vulnerability to genetic perturbations." However, it is entirely unclear what aspect of exposure to dietary choice is actually affecting mortality. Lyu et al. eLife 2021 demonstrated that flies on a choice diet show indicators of stress response, including increased locomotion and increased alcohol consumption. Additionally, meal choice potentiates the TCA cycle including elevating levels of alpha-ketoglutarate. The eLife paper showed that putatively normalizing levels of alpha-ketoglutarate through GDH knockdown abrogated the effect of a choice diet on lifespan, and could be directly involved in the effect of dietary choice on lifespan. There is no way to determine from the experiment in Fig. 6 whether diet-induced effects on metabolic network or perturbed alpha-ketoglutarate homeostasis is the key factor affecting starvation vulnerability in choice vs. fixed diet flies. Further, the effects of GDH and GS2 knockdown on starvation sensitivity are clear, but the effect of GS1 knockdown is not significant. In the same vein, the introduction and line 334 offer the speculation that changes in hub metabolites such as alpha-ketoglutarate may perturb network structure. However, this is not directly assessed in the experiments shown in Fig. 6.

Minor Critiques

- There are only 7 pathways with >6 metabolites. Is this standard coverage for fly metabolomics? It seems like low recovery.
- While 5-HT2A^{-/-} flies had less change in network structure on a choice diet, the 5-HT2A fixed diet network structure still differs from the WT fixed diet network, albeit with similar numbers of edges. Can statistics be computed for differences shown in panels 2C and 2D? Can you discuss the implications of having similar network connectivity metrics for a slightly different absolute network structure?
- Line 454 typo: interconnected
- Supplementary figures lack legends
- Figure 1 is so nicely organized and aligned and color coded! Beautifully clear.
- Figure 2A legend could use more exposition to make it clear to a non-expert audience.
- Figure 2E legend "p-value is marginal in bodies" - I would argue that a p-value of 0.1 is not marginal, it's non-significant.
- Color coding in Fig. 3 is confusing. In panels A and B, red vs. blue indicates node status. But in Panel C blue indicates 5-HT2A mutant. Consider differentiating these colors.
- Figure 4 was visually intimidating on first glance, but was generally well explained in the text. This is a minor point, but I did need to reread several times to understand the meaning of the colored bars identifying major groups along the margins. This should be described in the legend.
- Figure 5 – the font color difference between isovalerate and glutarate is so slight I can barely differentiate them and at first tried to find a connection between them, as choline is color matched between panels C and E. Consider recoloring one of them. Also, all of figure 5 has pastel background colors, which is likely an export artifact.

Reviewer #1 (Remarks to the Author):

The authors present evidence that diet-dependent changes in connectivity, average shortest distance, module clustering, and robustness are mediated by serotonin signaling through the 5-HT_{2A} receptor.

The work is well structured and the conclusions are supported from the original observations. I find the work overall novel, relevant to the aging/endocrine field and I recommend that it is published as is with no further revisions.

Reviewer #2 (Remarks to the Author):

Lyu, Promislow, and Pletcher perform additional analyses and computational network modeling on previously published metabolomics data (Lyu et al. 2021. *Elife* 10, e59399) and find that measures of network integrity decrease with dietary choice in a 5-HT_{2A} serotonin receptor dependent manner. In this manuscript, rather than focusing on specific metabolites or pathways, the authors consider metabolic networks as a whole and hypothesize that metabolic network structure itself may modulate lifespan. In accordance with data from Lyu et al. *eLife* 2021, loss of 5-HT_{2A} rescued network remodeling by the choice diet. The authors propose a model in which choice diet-induced metabolite network fragmentation would increase organismal vulnerability to perturbation. To test this model, they place flies on a choice diet to reduce network connectivity, add a “genetic perturbation” by knocking down enzymes that will affect alpha ketoglutarate levels, and finally assess starvation sensitivity. In accordance with their model, RNAi flies fed a choice diet had reduced starvation resistance compared with both WT flies and genetical control flies on a fixed diet. Experimentally linking disruption of metabolic networks to lifespan and mortality would be of great interest to many readers, but additional experimentation would be required to demonstrate this unequivocally.

Major comments

The network analysis was fascinating, as was the nearly complete rescue of the effects of choice diet on network structure by 5-HT_{2A} knockout. The major hypothesis laid out in the introduction that metabolic network fragmentation itself may regulate lifespan is intriguing. I have two major critiques. First, the *eLife* 2021 paper noted interesting effects of dietary switching between fixed and choice diets on lifespan, which were not well addressed in this manuscript. More significantly, the hypothesis testing experiment in Fig. 6 was ultimately unsatisfying in addressing the major questions posed by the authors.

1-1) In Fig. 1, the effects of dietary switching on lifespan are noted and metabolite correlations are shown for FD>CD and CD>FD. The effect of dietary switching on metabolite correlation is notable in heads (1F, 1G). In FD>CD heads (1F), there's a small increase in metabolite correlation vs. choice diet – similar to the differences in 1E, the implications of which aren't discussed. Even more confusingly, in CD>FD heads, there seems to be an increase in the most extreme correlations on FD, but the second-to-extreme bin in the CD>FD flies. Is this really meaningful change at all? The only mention of these panels is in lines 130-131.

We appreciate these comments from the reviewer as we struggled to find an effective way to represent and interpret the broad changes in correlation distributions among these treatments. We interpret the

differences in the switch treatments (e.g., 1F, 1G) as representing mixed distributions due to a partial reversal of the metabolite network structure 48 hours after the switch. We focused on emphasizing the extreme tails of the distributions because we found those differences were statistically significant (i.e., $FDR \leq 0.1$, see Supplementary Fig. 1), and we suggest that these highly-correlated metabolites form an important, stable component of the network. Whether they are meaningful is a question we can't answer definitively with our data set alone. We can say that they are influenced by 5-HT2A mutation, which is consistent with a role in influencing mortality.

We have made the following changes to clarify these points in the revision:

“We focused on the extreme tails of the correlation distribution. Visual inspection of the distribution suggested that while the mean across all correlations was not different, the tails of the distribution were considerably larger for fixed diet versus choice diet flies.” (Line 131-134)

“This partial reverse suggests a 48hr exposure to a new diet is sufficient to influence metabolite correlations, yet a complete transition (if possible) may take longer period of time as indicated by the mortality curves (Fig. 1b and 1c).” (Line 139-142)

1-2) When it comes to network construction (Fig. 2 B, Fig. S1) these switching studies are left out entirely, though edge analysis for these conditions is shown in Figs. 2C and 2D.

We apologize for omitting these panels, which are now included in the supplementary figures.

1-3) Beginning on line 204, you state, “Indeed, dietary switches, which led to rapid changes in mortality, also stimulated a 48hr modification of metabolic network edge number in the direction that reflected the influence of the new diet. These changes were observed in both heads and bodies (depicted as the 3rd and 4th columns in Figure 2C and 2D), and they mirror the changes in mortality.” However, particularly in heads, the FD>CD and CD>FD observed edges are nearly identical to one another. 48 hours is not a long time to remodel metabolic networks – these networks are likely still in flux toward their final state. Given the focus of the rest of the manuscript on serotonin receptor effects on network connectivity, the inclusion of these seemed like an underanalyzed afterthought that needs more consideration and interpretation.

In line with this reviewer's previous comment and our response to it, our goal here was to describe what we interpret as evidence of a partial reversal of the network structure in the switch treatments by noting the intermediate nature of the switch treatments relative to the unswitched controls. The observed edge numbers of FD->CD and CD->FD, for example, lie in between those observed in FD and CD, suggesting an ongoing transition. We have revised the text to reflect this point and to reinforce the trend suggested by Fig 1F, 1G.

“As also indicated by the correlation distributions, dietary switches, which led to rapid changes in mortality, also stimulated a 48hr modification of metabolic network edge number in the direction that reflected the influence of the new diet. The observed edge numbers of switched groups (depicted as the 3rd and 4th data groups in Fig. 2c and 2d), are intermediate to that of the unswitched groups (depicted

as the 1st and 2nd columns in Fig. 2c and 2d), supporting the notion of an ongoing transition in metabolic network structures 48hrs after dietary switch.” (Line 216-223)

We also recognize this reviewer’s point that the dietary switch data are valuable and may be used to understand and strengthen the links between metabolic network structures and mortality. In fact, our additional analyses do show temporal effects of switching on connectivity and community. Frankly, we were originally concerned that adding this would detract from our preferred focus on the influence of 5-HT2A/serotonin signaling on network properties, but considering this recommendation we are happy to add results about the switch networks (in Fig. 3a and Supplementary Fig. 4), which demonstrate the temporal nature of diet-induced network modulation.

Line 249-251, about connectivity: “Interestingly, switching to a fixed diet for 48hrs is sufficient to restore high-degree nodes, while switching to a choice diet for the same amount of time cannot eliminate those nodes (Fig. 3a, second row).”

Line 290-294, about community structure: “Dietary effects in module numbers were partially reversed by switching to another diet. Switching to a choice diet for 48hrs increased the module number (from 7 to 11, Supplementary Fig. 5a), while switching to a fixed diet for the same amount of time dramatically decreased the module number (from 31 to 12, Supplementary Fig. 5b).”

2-1) The effects of food choice on network fragmentation in clearly presented with multiple analyses that were well described to this non-metabolomic-modeling reader! However, the broad hypothesis from the introduction that metabolic network fragmentation itself may regulate lifespan is not actually tested. I concur with the legend for Fig. 6, which states that, “Exposure to dietary choice increases vulnerability to genetic perturbations.” However, it is entirely unclear what aspect of exposure to dietary choice is actually affecting mortality. Lyu et al. eLife 2021 demonstrated that flies on a choice diet show indicators of stress response, including increased locomotion and increased alcohol consumption. Additionally, meal choice potentiates the TCA cycle including elevating levels of alpha-

ketoglutarate. The eLife paper showed that putatively normalizing levels of alpha-ketoglutarate through GDH knockdown abrogated the effect of a choice diet on lifespan, and could be directly involved in the effect of dietary choice on lifespan. There is no way to determine from the experiment in Fig. 6 whether diet-induced effects on metabolic network or perturbed alpha-ketoglutarate homeostasis is the key factor affecting starvation vulnerability in choice vs. fixed diet flies.

We thank the reviewer for carefully considering one of the main conjectures that result from the work presented in our manuscript, that network fragmentation per se may influence lifespan. Recognizing that network structures (the study focus of this paper) and individual metabolites (as described in our eLife paper) are influenced by serotonin signaling and associated with mortality, we avoid asserting that one is more important than the other in modulating aging. Rather, we sought to advance previous inference from genetic analysis with a systems-biology perspective to generate testable hypotheses that might better distinguish cause and effect of these two factors. For example, as a first attempt to bridge the gaps between network structure, physiological robustness, and mortality, we proposed and tested an experimental starvation/genetic perturbation paradigm in Fig. 6. While our data are consistent with such a model, we recognize that they are not sufficient to infer fragmentation causes lifespan-shortening. Such experiments are well-aligned with our future research plans, but they involve large, complicated experiments that disrupt metabolic networks in such a way as to isolate the effects that caused by network changes from those caused by changes in individual molecules. We therefore respectfully argue that such experiments lie beyond the scope of the current manuscript and suggest the following textual revisions to address this:

Line 362-366: “Our systems analyses motivated us to ~~test~~ propose a working mechanistic model (Fig. 6a) in which the effects of dietary choice on network structure, which include increased fragmentation as well as reduced connectivity, average shortest distance, and robustness, would increase organism-level vulnerability to genetic or environmental perturbations, akin to removal of network nodes, as measured by resistance to stress. “

Line 386-389, we added: “In summary, our data are consistent with the predicted model (Fig. 6a). Whether such influences are due to changes in network structures or in individual metabolites (e.g. α -ketoglutarate, as indicated by our previous study³¹), remains to be determined.”

Line 438-440, we revised our discussion into: “Nevertheless, the preservation of network structure alone may not provide a complete picture of the mechanisms that influence physiological robustness, as the manifestation of biological robustness occurs on many different levels. To better understand these systems, future experiments should attempt to isolate effects caused by network change from those caused by changes in individual molecules.”

Finally, recognizing the complexity of aging, the field has, to some extent, embraced systems biology, but, in our opinion, most efforts have focused largely on descriptive experiments and have lacked genetic analysis, model prediction, and subsequent hypothesis testing that we strived to develop in this work.

2-2) Further, the effects of GDH and GS2 knockdown on starvation sensitivity are clear, but the effect of GS1 knockdown is not significant.

We clarified the statistics and inference as suggested. We maintained the conclusion though as the trend is consistent among the three enzyme knock-downs.

“Importantly, we observed a consistent trend that knocking down any one of the three enzymes exacerbated the differences between a choice and a fixed diet, as predicted by our model (Fig. 6d-6f; but note that the P -value associated with GS1 knockdown does not reach significance, $P(\text{interaction}) = 0.13$).” (Line 381-384)

2-3) In the same vein, the introduction and line 334 offer the speculation that changes in hub metabolites such as alpha-ketoglutarate may perturb network structure. However, this is not directly assessed in the experiments shown in Fig. 6.

As discussed above, this point is well-taken, as our conjecture was based on the results in this manuscript and our previously published paper. But follow-up experiments to establish this inference are large and expensive. They are, in fact, a primary interest for the first author when she begins her independent laboratory. We therefore respectfully argue that they are beyond the scope of the current contribution.

Minor Critiques

1) There are only 7 pathways with >6 metabolites. Is this standard coverage for fly metabolomics? It seems like low recovery.

Coverage depends on the number of detectable metabolites and which metabolites are targeted in the metabolomic panels, and it would be hard to compare between studies with different designs. In our study we have detected 103 head metabolites and 125 body metabolites, and the seven pathways with the >6 metabolites cut-off recovered large number of metabolites in our dataset.

2) While 5-HT2A^{-/-} flies had less change in network structure on a choice diet, the 5-HT2A fixed diet network structure still differs from the WT fixed diet network, albeit with similar numbers of edges. Can statistics be computed for differences shown in panels 2C and 2D? Can you discuss the implications of having similar network connectivity metrics for a slightly different absolute network structure?

We chose not to focus on the effects of loss of 5-HT2A in flies fed a fixed diet because of the mutations relatively small effect in this treatment. While the reviewer makes a strong point that the fixed diet data indicate connectivity and community are not always affected similarly, with the latter perhaps more influential on mortality. As a result we added the statistics for genotype differences for each dietary group (as shown in the new Supplementary Fig. 3), and we discussed their implications.

Line 229-234: “We found that the diet effect was reduced in 5-HT2A mutant flies (Fig. 2c and 2d) compared to control animals, primarily due to more edges in the network constructed from the heads of flies that fed on a choice diet (Supplementary Fig. 3, permutation test, $P=0.0022$). When fed on a fixed diet, genotype differences were not significant (Supplementary Fig. 3, permutation test, $P=0.30$ and $P=0.40$ in heads and bodies respectively).”

Line 311-319: “Correlation plots revealed that dietary influences on the number and size of modules were largely abrogated in the heads of *5-HT2A* mutants (Fig. 4b), as these flies exhibited decreased/increased module numbers when fed on a choice or a fixed diet, respectively. Notice for mutants that fed a fixed diet, despite their network connectivity remained the same as control flies on this diet (Fig. 3a and Supplementary Fig. 3), their community structures showed differences. Such changes are associated with the increased mortality of *5-HT2A* mutants on a fixed diet (Fig. 1a and 1d), suggesting network fragmentation may have more implications in aging than overall connectivity.”

(New supplementary Fig. 3)

3) Line 454 typo: interconnected.

We fixed the typo. Thank you for your careful reading.

4) Supplementary figures lack legends.

We apologize for omitting the legends, which are now included in the revision.

5) Figure 1 is so nicely organized and aligned and color coded! Beautifully clear.

Thank you!

6) Figure 2A legend could use more exposition to make it clear to a non-expert audience.

We greatly expanded the legend of Figure 2 to improve readability.

7) Figure 2E legend “p-value is marginal in bodies” - I would argue that a p-value of 0.1 is not marginal, it’s non-significant.

We changed this statement as follow:

“Such trend is also observed in bodies (f), yet the *P*-value is not significant ($P=0.1025$, permutation test).”

8) Color coding in Fig. 3 is confusing. In panels A and B, red vs. blue indicates node status. But in Panel C blue indicates *5-HT2A* mutant. Consider differentiating these colors.

Thanks for the advice. We now use grey instead of blue to represent node status.

9) Figure 4 was visually intimidating on first glance, but was generally well explained in the text. This is a minor point, but I did need to reread several times to understand the meaning of the colored bars identifying major groups along the margins. This should be described in the legend.

We expanded the description on the module colors in the legend.

10) Figure 5 – the font color difference between isovalerate and glutarate is so slight I can barely differentiate them and at first tried to find a connection between them, as choline is color matched between panels C and E. Consider recoloring one of them. Also, all of figure 5 has pastel background colors, which is likely an export artifact.

We changed the colors for isovalerate and glutarate to improve visibility. The pastel background was removed upon suggestion.

REVIEWERS' COMMENTS:

Reviewer #2 (Remarks to the Author):

I appreciate the authors' detailed response. I believe the softened language about the need for further followup is appropriate and appreciate the inclusion of additional detail in various sections of the manuscript and legends. I certainly look forward to future studies to further dissect the consequences of network disruption.

I recommend acceptance of this manuscript.